# Enhanced Expression of ARK5 in Hepatic Stellate Cell and Hepatocyte Synergistically Promote Liver Fibrosis

**DOI:** 10.3390/ijms232113084

**Published:** 2022-10-28

**Authors:** Yang You, Chongqing Gao, Junru Wu, Hengdong Qu, Yang Xiao, Ziwei Kang, Jinying Li, Jian Hong

**Affiliations:** 1Department of Pathophysiology, School of Medicine, Jinan University, Guangzhou 510630, China; 2Department of Hepatological Surgery, The First Affiliated Hospital, Jinan University, Guangzhou 510630, China; 3Department of Gastroenterology, The First Affiliated Hospital, Jinan University, Guangzhou 510630, China

**Keywords:** ARK5, hepatic stellate cell, hepatocyte, TGF-β signaling pathway, liver fibrosis

## Abstract

AMPK-related protein kinase 5 (ARK5) is involved in a broad spectrum of physiological and cell events, and aberrant expression of ARK5 has been observed in a wide variety of solid tumors, including liver cancer. However, the role of ARK5 in liver fibrosis remains largely unexplored. We found that ARK5 expression was elevated in mouse fibrotic livers, and showed a positive correlation with the progression of liver fibrosis. ARK5 was highly expressed not only in activated hepatic stellate cells (HSCs), but also in hepatocytes. In HSCs, ARK5 prevents the degradation of transforming growth factor β type I receptor (TβRI) and mothers against decapentaplegic homolog 4 (Smad4) proteins by inhibiting the expression of Smad ubiquitin regulatory factor 2 (Smurf2), thus maintaining the continuous transduction of the transforming growth factor β (TGF-β) signaling pathway, which is essential for cell activation, proliferation and survival. In hepatocytes, ARK5 induces the occurrence of epithelial-mesenchymal transition (EMT), and also promotes the secretion of inflammatory factors. Inflammatory factors, in turn, further enhance the activation of HSCs and deepen the degree of liver fibrosis. Notably, we demonstrated in a mouse model that targeting ARK5 with the selective inhibitor HTH-01-015 attenuates CCl_4_-induced liver fibrosis in mice. Taken together, the results indicate that ARK5 is a critical driver of liver fibrosis, and promotes liver fibrosis by synergy between HSCs and hepatocytes.

## 1. Introduction

Liver fibrosis is a wound healing response, triggered by a variety of chronic liver diseases [1]. It is characterized by the proliferation of myofibroblasts, hepatocyte damage, and excessive extracellular matrix (ECM) deposition, which eventually results in organ dysfunction and death [2,3]. However, no effective liver-specific antifibrosis drugs have been developed to date. Therefore, a better understanding of the molecular mechanism of liver fibrosis would facilitate the development of preventive and therapeutic approaches for liver fibrosis.

Activated hepatic stellate cells (aHSCs) play a critical role in liver fibrogenesis, as a main source of myofibroblasts and a major contributor to the ECM [4,5]. TGF-β, one of the most potent inducers of fibrogenesis, is believed to play a key role in transforming quiescent HSCs into fibrogenic myofibroblasts [6]. Additionally, aHSCs can release profibrotic cytokines, including TGF-β, to promote liver fibrosis [6,7]. The positive feedback between HSCs and their microenvironment contributes to the persistent activation of HSCs [7].

Hepatocyte damage or death is the key trigger of acute and chronic liver disorders [8,9]. Recent studies have shed light on the profibrogenic roles of injured hepatocytes [8,10,11]. EMT is a critical process for an epithelial cell to undergo a conversion to a mesenchymal phenotype, and is believed to be an inflammation-induced response [12]. Indeed, hepatocyte EMT is another important source of fibroblasts that promotes collagen deposition in the liver [13]. Moreover, the crosstalk between hepatocytes and HSCs plays a key role in the progression of fibrosis. Biomolecules from injured hepatocytes can contribute to the initiation of HSCs activation [8,11]. However, the precise mechanisms underlying the regulation functions of hepatocytes during the initiation and progression of liver fibrosis have not been fully understood.

ARK5 is a serine/threonine kinase and has been identified as an essential part of the catalytic subunit of the AMPK (AMP-activated protein kinase) family [14]. Previous studies have found that ARK5 participates in a number of significant functions such as cell adhesion, cell proliferation and metastasis, and carbohydrate metabolism [15]. Aberrant expression of the ARK5 has been observed in a wide variety of solid tumors, including liver cancer, and positively correlates with the malignancy of cancer cells [16,17,18]. Recently, ARK5 has been identified as a gene that is strongly associated with fibrosis, TGF-β signaling, and poor outcomes in human chronic kidney disease [14,19]. However, the expression and TGF-β signaling interactions of the ARK5 during the development of liver fibrosis remains to be elucidated.

The TGF-β pathway begins with the activation of TGF-β ligands and their binding to the TβRII [20]. TβRII then recruits and phosphorylates TβRI, which in turn phosphorylates downstream Smad2/3 [21]. Phosphorylated Samd2/3 subsequently recruits Smad4, translocates to the nucleus and regulates the transcription of various genes [22]. Post-translational modifications are an important mechanism regulating the expression, localization, and activity of TGF-β signaling components [23]. In particular, ubiquitination-mediated protein degradation has gradually become a key pathway in post-transcriptional translation, regulating the intensity and duration of intracellular TGF-β signaling through the actions of various E3 ubiquitin ligases and deubiquitinases [23,24].

Here, we showed that ARK5 expression is elevated in HSCs and hepatocytes during the progression of liver fibrosis. ARK5 accumulation promotes the activation and proliferation of HSCs, through preventing the degradation of TβRI and Smad4 by inhibiting the expression of E3 ubiquitin ligase Smurf2, thereby maintaining the activity of the TGF-β pathway. Moreover, ARK5 induced EMT and inflammatory factors secretion in hepatocytes, further aggravating the degree of liver fibrosis. Together, these findings highlight a profibrotic role of ARK5 and revealed the mechanism by which ARK5 in HSCs and hepatocytes synergistically promotes liver fibrosis. Targeting ARK5 may be an effective therapeutic strategy for liver fibrosis.

## 2. Results

### 2.1. ARK5 Expression Level Correlates with the Progression of Liver Fibrosis, and Mainly Expresses in HSCs and Hepatocytes

To examine whether ARK5 expression is altered in the fibrotic livers, a CCl_4_-induced mouse model of liver fibrosis was established. The results of Western blot showed that the level of ARK5 protein was significantly enhanced in mouse fibrotic liver tissues (Figure 1A). Moreover, we also performed an immunohistochemistry analysis of ARK5 and Sirius red staining of mouse fibrotic liver sections at different stages. The staining area of ARK5 gradually increased from 0 weeks to 8 weeks, which was consistent with the increasing trend of the Sirius red staining area (Figure 1B,C), suggesting that hepatic ARK5 might be positively related to the degree of liver fibrosis. To further clarify the cellular origin of ARK5, histological staining and immunohistochemistry were used to detect the localization of ARK5 in mouse fibrotic liver tissues. We found that ARK5 expression largely co-localized with fibrotic deposition and the HSCs activation marker alpha smooth muscle actin (α-SMA), suggesting that ARK5 was mainly increased in activated HSCs. Interestingly, ARK5 expression was also elevated in surrounding hepatocytes and co-localized with α-SMA staining (Figure 1D). To further validate our findings, we investigated the ARK5 expression in primary HSCs and hepatocytes isolated from mouse normal and fibrotic livers. As expected, the mRNA and protein levels of ARK5 were significantly increased in HSCs and hepatocytes from fibrotic mouse livers (Figure 1E). Taken together, these results indicated that ARK5 expression was positively correlated with the severity of liver fibrosis, and was mainly contributed to by active HSCs and hepatocytes.

### 2.2. ARK5 Plays a Crucial Role in the Activation, Proliferation and Survival of HSCs

To further explore the regulatory role of ARK5 in HSCs, we knocked down ARK5 using siRNA in LX-2 cells. Quantitative PCR data showed that the mRNA level of HSC activation genes *actin alpha 2* (*ACTA2*), *platelet derived growth factor receptor-β* (*PDGFRβ*), and pro-fibrosis markers *collagen 1A1* (*COL1A1*), *plasminogen activator inhibitor 1* (*PAI-1*) and *tissue inhibitor of metalloproteinase 1* (*TIMP1*) were remarkably diminished in ARK5-knockdown LX-2 cells (Figure 2A). Consistently, protein levels of α-SMA and Collagen I were down-regulated (Figure 2B). In addition, the results of wound healing assay and Transwell assay showed that knockdown of ARK5 inhibited the migration ability of LX-2 (Figure 2C,D). These results also suggest that the knockdown of ARK5 may affect the proliferation and survival of LX-2 cells. Subsequently, CCK8 assay and flow cytometry were used for further verification. As expected, the knockdown of ARK5 inhibited the proliferation of LX-2 cells and also triggered some cell apoptosis (Figure 2E,F). Taken together, these findings indicated that ARK5 promoted HSCs activation and proliferation, as well as maintained the survival of HSCs.

### 2.3. ARK5 Promotes HSCs Activation via TGFβ-Smad2/3 Activity

During acute and chronic liver injury, HSCs can be induced to activate and transdifferentiate to myofibroblasts by TGF-β1. We observed that TGF-β1 strongly induced ARK5 in LX-2 (Appendix A). In addition, TGFβ receptor I (TβRI) inhibitor SB431542 decreased ARK5 expression induced by TGF-β1 in LX-2 (Appendix A), indicating that ARK5 was a TGF-β1 responsive gene. 

To investigate the functional impact of ARK5 in aHSCs, we knocked down the expression of ARK5 in LX-2 cells under TGF-β1 stimulation. Our data showed that ARK5 knockdown markedly suppressed the mRNA and protein levels of α-SMA and PAI-1 (Figure 3A). Therefore, we hypothesized that ARK5 activated HSCs by regulating the TGF-β pathway. As expected, the knockdown of ARK5 reduced the TGF-β-stimulated Smad2/3 phosphorylation (Figure 3B). To further examine the regulation of Smad2/3 phosphorylation by ARK5, we also performed immunofluorescence staining. After TGF-β1 stimulation, phosphorylated Smad2/3 mostly translocated to nucleus in LX-2 cells transfected with siControl, whereas phosphorylated Smad2/3 only partially localized to the nucleus in LX-2 cells transfected with siARK5 (Figure 3C). As TβRI mediates phosphorylation of Smad2/3, while Smad4 mediates nuclear translocation of p-Smad2/3. We further analyzed whether ARK5 could regulate TβRI and Smad4 during liver fibrosis. We found that ARK5 knockdown decreased the protein levels of TβRI and Smad4 in LX-2 cells (Figure 3D). In summary, we demonstrated that TGF-β1 induces a rapid increase ARK5 in HSCs. ARK5, in turn, can promote TGF-β-Smad2/3 signaling, which forms a positive feedback loop, ultimately leading to liver fibrosis.

### 2.4. ARK5 Enhances TβRI and Smad4 Protein Stability by Suppressing Its Ubiquitination

To test whether ARK5 regulates TβRI and Smad4 at the transcriptional level, we used qPCR for further validation. Interestingly, the mRNA level of both *TβRI* and *S**MAD4* was not altered after knockdown of ARK5 expression in LX-2 cells (Figure 4A). These results indicated that ARK5 may regulate TβRI and Smad4 at the post-transcriptional level. Therefore, we examined the degradation of TβRI and Smad4 proteins in siControl and ARK5-knockdown LX-2 cells after treatment of cells with cycloheximide (CHX). We found that knockdown of ARK5 promoted the degradation of TβRI and Smad4 (Figure 4B). However, proteasome inhibitor MG132 treatment reversed the downregulated of TβRI and Smad4 proteins by ARK5 knockdown in LX-2 cells (Figure 4C), suggesting that ARK5 might stabilize TβRI and Smad4 proteins through inhibiting proteasome-mediated degradation. To further validate this finding, we performed a ubiquitination assay in LX-2 cells and found that the knockdown of ARK5 increased the ubiquitination of TβRI and Smad4 (Figure 4D). Taken together, these findings suggested that ARK5 protected TβRI and Smad4 from ubiquitination and proteasome degradation.

### 2.5. ARK5 Reduces the Degradation of TβRI and Smad4 by Inhibiting the Expression of Smurf2

The activity of TGF-β/Smad pathway is known to be regulated by ubiquitination at multiple levels [25]. As E3 ubiquitin ligases, Smad ubiquitination regulatory factors (Smurfs) play an important role in mediating the ubiquitination and degradation of TGF-β/Smad pathway proteins [26]. Therefore, we speculated that ARK5 regulates TβRI and Smad4 degradation by modulating the expression of Smurfs. The results of qPCR showed that the expression of *SMURF2*, but not *SMURF1*, was up-regulated after the knockdown of ARK5 in LX-2 cells (Figure 5A). Consistently, this result was further confirmed by Western blot (Figure 5B). Next, we explored the binding of Smurf2 to TβRI and Smad4 in LX-2 cells. Co-immunoprecipitation results showed that the interaction of Smurf2 with TβRI and Smad4 was increased in ARK5-knockdown cells (Figure 5C). To further investigate the role of Smurf2 in the degradation of TβRI and Smad4, we used siRNA to interfere with the expression of Smurf2 in ARK5-knockdown LX-2 cells. The results showed that the expression of TβRI and Smad4 was restored in ARK5-knockdown LX-2 cells co-transfection with siSmurf2 (Figure 5D–F). Taken together, these findings indicated that ARK5 maintains TGF-β/Smad pathway activation by inhibiting Smurf2-mediated degradation of TβRI and Smad4 proteins in HSCs.

### 2.6. Enhanced Expression of ARK5 Promotes the Hepatocyte EMT during Liver Fibrosis

Previous studies have shown that ARK5 is involved in the regulation of EMT processes in a variety of tumors [14]. Therefore, we speculate that ARK5 has a regulatory role in hepatocyte EMT during liver fibrosis. Hepatocyte nuclear factor alpha (HNF-4α) (hepatocyte marker), α-SMA (mesenchymal marker), and ARK5 protein levels were detected in liver tissues derived from mice treated with CCl_4_ and controls. Immunofluorescence results showed that, in mouse fibrotic liver tissues, some ARK5-positive hepatocytes were HNF-4α-negative, while some ARK5-positive hepatocytes were α-SMA-positive (Figure 6A). These suggested that the increased expression of ARK5 was closely related to the EMT of hepatocytes. TGF-β1 is a potent inducer of fibrogenic EMT [27], therefore we transfected human hepatic LO2 cells with ARK5 siRNA and treated with TGF-β1 to further clarify the role of ARK5 in hepatocyte EMT. We found that ARK5 knockdown reversed the increasing of the α-SMA and Vimentin induced by TGF-β1 and maintained E-cadherin levels (Figure 6B). Immunofluorescence co-staining also showed that the number of α-SMA positive cells was reduced in siARK5-transfected TGF-β1-treated LO2 cells compared with siControl-transfected TGF-β1-treated LO2 cells (Figure 6C). In summary, we demonstrated that increased expression of ARK5 promotes EMT in hepatocytes during liver fibrosis.

### 2.7. Hepatocyte ARK5 Mediates the Activation of HSCs by Regulating the Release of Inflammatory Factors

To explore whether hepatocyte ARK5 could regulate the HSCs activation, we used hepatocyte conditioned medium (CM) for further studies. LO2 cells were transfected with ARK5 overexpression plasmid; the supernatant was collected after 24 h for the culture of LX-2 cells (Figure 7A). We found that the protein expression level of ARK5 was increased in LO2 transfected with ARK5 overexpression plasmid (Figure 7B). Then, we investigated the activation of LX-2 cultured with CM. The expression of ARK5 and HSC activation genes *ACTA2*, *PDGFRβ*, *COL1A1*, *PAI-1* and *TIMP1* was elevated in LX-2 treated with CM from hepatocytes overexpressing ARK5 (Figure 7C). To further clarify the regulatory mechanism of hepatocyte ARK5 on HSCs activation, we detected the level of inflammatory factors in hepatocytes. qPCR analyses showed that the mRNA level of *IL-6*, *IL-18* and *TGF-β1* was up-regulated in the ARK5-overexpressing LO2 cells (Figure 7D). This result was further confirmed by ELISA assays (Figure 7E). Taken together, these findings suggested that hepatocyte ARK5 promotes HSCs activation and liver fibrosis by regulating the expression and release of multiple inflammatory factors.

### 2.8. ARK5 Inhibitor Treatment Attenuates Liver Fibrosis in Mice

To explore whether targeting ARK5 could prevent further progression of established fibrosis in vivo, we used ARK5-specific inhibitor HTH-01-015 to treat CCl_4_-induced mouse fibrosis models (Figure 8A). The inhibitor HTH-01-015 was divided into two concentrations. Histological staining showed that mice in the HTH-01-015 treated groups had less liver damage and fewer Sirius red stained areas compared to the vehicle control group (Figure 8B and Appendix A). Correlated with the results of Sirius red staining, the immunohistochemical staining areas of α-SMA, Collagen I and ARK5 were decreased in the liver tissue of HTH-01-015-treated mice compared with the vehicle control group (Figure 8B and Appendix A). These results showed that the activation of HSCs and the EMT of hepatocytes were inhibited in mice. In addition, compared with the vehicle control group, the levels of hepatic hydroxyproline and serum alanine aminotransferase (ALT) and aspartate aminotransferase (AST) were down-regulated in the HTH-01-015 treatment groups (Figure 8C,D). Western blot further confirmed that HTH-01-015 could reduce the protein levels of fibrosis markers (α-SMA, Collagen I and PAI-1) in liver tissues (Figure 8E). It is worth noting that the treatment effect of the high-dose group was better in the above various tests. Taken together, these results showed that the inhibition of ARK5 effectively prevents the progression of liver fibrosis in mice.

## 3. Discussion

It is known that liver fibrosis is a persistent wound-healing response caused by a variety of chronic liver diseases, characterized by the production of a large amount of ECM [28]. If unchecked, liver fibrosis will gradually develop into liver cirrhosis, hepatocellular carcinoma, and liver failure [29]. ARK5 is an important member of the AMPK family and is involved in complex biological operations such as cell migration, invasion, adhesion, proliferation, metabolism, senescence, apoptosis and survival [15]. Studies have shown that ARK5 is highly expressed in hepatocellular carcinoma and is associated with poor tumor prognosis [30,31]. However, the role and mechanism of ARK5 in the process of liver fibrosis are rarely reported. In this work, we have confirmed that ARK5 plays a critical role in the development of liver fibrosis. We found that ARK5 expression was up-regulated in mouse fibrotic liver tissues compared to normal mouse liver tissues. In addition, we found that the expression of ARK5 was correlated with the degree of liver fibrosis. Meanwhile, we found that ARK5 was mainly highly expressed in HSCs and hepatocytes in fibrotic livers. This suggests that the ARK5 in HSCs and hepatocytes may play an important role in promoting the development of liver fibrosis.

Myofibroblasts come from a variety of cellular sources, including epithelial cells, mesenchymal stromal cells, fibroblasts, mesothelial cells, HSCs, and portal fibroblasts. Among them, HSCs are the main precursors of myofibroblasts [32]. Upon stimulation of liver injury, quiescent hepatic stellate cells (qHSCs) are activated and transdifferentiated into myofibroblast-like cells that actively regulate tissue repair [33]. Compared with qHSCs, activated HSCs have novel properties such as proliferation, contraction, enhanced ECM synthesis, chemotaxis, and inflammatory signaling, and was accompanied by changes in transcriptional signatures, including increased expression of α-SMA and collagen I [34]. The reduction of ECM contributes to the regression of fibrosis, thus targeting activated HSCs (including induction of apoptosis and inhibition of cell activation), and has become the main anti-fibrotic strategy [35]. We used siRNA to interfere with the expression of ARK5 in LX-2 to explore the function of ARK5 in HSCs. Findings showed that knockdown of ARK5 can down-regulate the expression of activation-related and fibrosis-related marker genes in HSCs. In addition, knockdown of ARK5 in HSCs can inhibit cell migration and proliferation, and even induce apoptosis. The above results show that ARK5 plays an important role in the activation, proliferation and survival of HSCs. Targeting ARK5 in hepatic stellate cells may be an effective strategy for the treatment of liver fibrosis.

Next, we explored the exact ARK5-associated regulatory mechanism in HSCs. Findings showed that ARK5 was a TGF-β pathway responsive gene. In addition, we found that knockdown of ARK5 inhibited TGF-β1-induced activation of HSCs. These results suggested that ARK5 may mediate the activation of HSCs by regulating the TGF-β pathway. Our further studies confirmed that reducing ARK5 expression inhibited TGF-β1 induced p-Smad2/3 expression and translocation to the nucleus. When the TGF-β pathway is activated, Smad2/3 proteins are phosphorylated by TβRI, and then p-Smad2/3 binds to Smad4 to form a heterogeneous complex, which is transported to the nucleus to interact with abundant transcription factors, regulating target genes transcription, translation, microRNA biogenesis, protein synthesis and post-translational modifications [36,37]. Therefore, we further explored the regulatory role of ARK5 in the expression of TβRI and Smad4. Interestingly, we found that knockdown of ARK5 down-regulated TβRI and Smad4 proteins, but rarely affected their mRNA levels. These results indicated that ARK5 may regulate TβRI and Smad4 through a post-translational manner. The level, localization, and function of TGF-β signaling mediators, regulators, and effectors are all affected by ubiquitination, a key post-transcriptional modification [23]. Our further studies confirmed that TβRI and Smad4 are indeed degraded by ubiquitination. Smurfs belong to the Nedd4 subfamily of HECT-type E3 ubiquitin ligases and are involved in the ubiquitination and degradation of various TGF-β pathway-related proteins [26]. Therefore, we next tested whether knockdown of ARK5 could affect the expression of Smurfs. The results of qPCR and Western blot showed that knockdown of ARK5 up-regulated the expression level of Smurf2, and the results of immunoprecipitation further showed that the binding of Smurf2 to TβRI and Smad4 was increased after knockdown of ARK5. In addition, Western blot and immunofluorescence analysis showed that interference Smurf2 expression in ARK5-knockdown LX-2 cells reduced the degradation of TβRI and Smad4. These results suggested that ARK5 mediates the degradation of TβRI and Smad4 by regulating the expression of Smurf2. Certainly, additional study will be needed to illustrate the detailed mechanism.

There is increasing evidence that hepatocytes play an important role in liver fibrosis. Here, we explored the function of ARK5 in hepatocytes. HNF-4α is a master regulator of hepatocyte phenotype and a key factor in xenobiotic metabolism, and reduced expression of HNF-4α has been reported in liver fibrosis [38]. We found that some ARK5-positive hepatocytes were α-SMA positive but HNF-4α negative in mouse fibrotic liver tissue, suggesting ARK5 overexpression was associated with hepatocyte EMT. The EMT is a morphogenetic altered and reversible process, during which an epithelial cell loses its own features by reducing some epithelial proteins expression (E-cadherin, β-catenin, and ZO-1) and by acquiring a mesenchymal phenotype, thereby increasing mesenchymal proteins expression (N-cadherin, Fibronectin, Vimentin and α-SMA) [39]. TGF-β1 is the main inducer of the EMT process [40]. Our in vitro results showed that silencing ARK5 in hepatocytes suppressed TGF-β1-induced upregulation of α-SMA and Vimentin expression, further suggesting that ARK5 mediates hepatocyte EMT during liver fibrosis. Cellular crosstalk between HSCs and surrounding cells contributes to the activation of HSCs and the progression of liver fibrosis [41]. Therefore, we next continued to explore whether the high expression of ARK5 in hepatocytes could affect the activation of HSCs. Our data show that the expression of HSCs activation-related genes is significantly upregulated in LX-2 cells after culture with medium from ARK5-overexpressing hepatocytes. In the following experiments, we found that the overexpression of ARK5 in hepatocytes caused changes in the expression of some inflammatory factors. Among them, the expressions of IL-6, IL-18 and TGF-β1 were significantly increased. Previous studies have shown that IL-6 activates HSCs by inducing the activation of the STAT3 pathway [42]. In addition, a recent study confirmed that IL-18 can directly promote the activation of hepatic stellate cells in mouse liver fibrosis [43]. Together with TGF-β1, they mediate the activation of HSCs through different mechanisms. Of course, ARK5 may also regulate the expression of many other inflammatory factors in hepatocytes, which may also play a role in HSCs activation, but it is not yet clear. Therefore, further studies are needed to clarify the regulatory role of ARK5 in hepatocytes.

Since ARK5 plays an important regulatory role in cells, abnormal expression of ARK5 often leads to cell dysfunction, which in turn induces a variety of diseases. At present, ARK5 has become a potential therapeutic target for various diseases such as neurodegenerative diseases and cancers and has received extensive attention [15]. The research and development of inhibitors against ARK5 and its isoforms has made great progress, and the specificity of the inhibitors has been continuously improved [15,44]. HTH-01-015 is a highly selective protein kinase inhibitor, which mainly affects ARK5 by inhibiting the phosphorylation of myosin phosphatase target subunit 1 [44]. Therefore, we chose it to treat CCl_4_-induced liver fibrosis in mice in the following experiments to determine whether targeting ARK5 could reverse liver fibrosis. As expected, HTH-01-015 reduced fibrosis deposition and the expression of fibrosis-related proteins such as α-SMA and Collagen I in liver tissues of fibrosis model mice. The area of ARK5 immunohistochemical staining decreased in the treatment group, further confirming our previous findings of the relationship between ARK5 and the degree of liver fibrosis. In addition, the liver function of the mice was also restored under the treatment of HTH-01-015. In particular, we found that the high-dose treatment group showed a better treatment effect compared to the low-dose treatment group. Our results confirm that ARK5 can serve as a potential target for liver fibrosis therapy. In addition, we also found that HTH-01-015 can also be used as a targeted drug against liver fibrosis, which has potential clinical application value. However, the pharmacological effects of this inhibitor in mice are not fully understood, and more in-depth and comprehensive exploration is needed.

## 4. Materials and Methods

### 4.1. Mouse Liver Fibrosis Models

Specific pathogen-free C57BL/6J mice were obtained from the Medical Laboratory Animal Center, Guangdong, China. They were housed under a controlled temperature (20 °C ± 2 °C) with 12-h light/12-h dark cycles, and were provided with free access to food and water. All animal studies were approved by the Animal Care and Use Committee at Jinan University.

Liver fibrosis was induced by gavage of 40% carbon tetrachloride (CCl_4_) in olive oil at 1 mL/kg body weight, twice per week for 8 weeks. Controls were only administered an equal volume of olive oil vehicle. Liver tissues were collected at 0 weeks, 2 weeks, 4 weeks, 6 weeks and 8 weeks. In the ARK5 inhibitor treatment model, liver fibrosis was induced in mice as described above, and HTH-01-015 (10 mg/kg or 20 mg/kg) was administered by intraperitoneal injection every other day over 5–8 weeks. The animals were sacrificed 3 days after the final treatment. The liver and serum were harvested for analysis.

### 4.2. Cell Culture and Treatments

LX-2 cells and LO2 cells were cultured in Dulbecco’s modified Eagle’s medium (DMEM; Gibco), supplemented with 10% fetal bovine serum (FBS; Hyclone) and 1% antibiotic (penicillin and streptomycin; Gibco) and incubated at 37 °C with 5% CO_2_. The short interfering RNAs (siRNA) were synthesized by Ribobio (Guangzhou, China). Transfection plasmids were constructed by GeneCopoeia (Guangzhou, China). Cells were transfected using lipofectamine 3000 (Invitrogen, Waltham, MA, USA) according to the manufacturer’s instructions. Cultured cells were treated with TGF-β1 (10 ng/mL), SB431542 (10 mM), Cycloheximide (50 μg/mL) or MG132 (20 μM) and harvested at the indicated time.

### 4.3. Isolation of Primary HSCs and Hepatocytes from Mouse Liver

Primary mouse HSCs and hepatocytes were isolated as previously described [45]. Briefly, the isolation of HSCs by stepwise retrograde perfusion digestion using a solution containing pronase (Sigma-Aldrich, St. Louis, MO, USA) and collagenase (Roche, Basel, Switzerland). The isolation of hepatocytes from mouse livers was carried out using a two-step collagenase perfusion method. HSCs and hepatocytes were purified with density gradient centrifugation and cultured in DMEM (containing 10% fetal bovine serum) and incubated at 37 °C with 5% CO_2_.

### 4.4. Histology and Immunostaining

Liver samples were fixed in 10% neutral formalin fixative and embedded in paraffin, and were sectioned at 4 µm thickness. Histological analysis was performed in hematoxylin-eosin staining and Sirius red staining. For immunohistochemistry, liver sections were incubated with anti-ARK5 (Santa, Santa Cruz, CA, USA, sc-271828), anti-α-SMA (Invitrogen, Waltham, MA, USA, PA5-18292), anti-collagen I (Invitrogen, Waltham, MA, USA, PA5-29569) primary antibodies overnight at 4 °C. Positive staining was visualized with the EnVision immunodetection system (K5007; Dako, Copenhagen, Denmark) according to the manufacturer’s instructions. For immunofluorescence, liver sections or cells were fixed in ice-cold methanol and were incubated with anti-ARK5 (Santa, Santa Cruz, CA, USA, sc-271828), anti-α-SMA (Invitrogen, Waltham, MA, USA, PA5-18292), anti-p-Smad2/3 (Abcam, Cambridge, UK, ab254407), anti-Smurf2 (Santa, Santa Cruz, CA, USA, sc-518164), anti-TβRI (Santa, Santa Cruz, CA, USA, sc-518018), anti-Smad4 (Cell Signaling Technology, Beverly, MA, USA, 46535S) and anti-HNF-4α (Abcepta, Suzhou, China, AP60014) antibodies followed by incubation with Alexa Fluor 488 or 555-conjugated secondary antibodies (Invitrogen, Waltham, MA, USA). Nucleis were counterstained with DAPI. Images were captured using an inverted fluorescence microscope or confocal laser scanning microscope.

### 4.5. Quantitative PCR Analysis

Total RNA was extracted from cells by using Trizol reagent (Thermo, Logan, UT, USA) and was reverse transcribed to cDNA using the PrimeScript 1st Strand cDNA Synthesis Kit (RR036A; TaKaRa, Shiga, Japan) according to the manufacturer’s suggestions. Quantitative PCR used SYBR Green qPCR Master mix (K1070; Apexbio, Houston, USA) on the CFX96 Real-Time system (Bio-Rad, Hercules, CA, USA) according to the manufacturer’s instructions. The results of the qPCR were standardized using *GAPDH* and analyzed by 2^−ΔΔCT^. The primers used for qPCR are listed as follows: *Ark5*: forward 5′-GATGGCCCTGCTGTAGAGAC-3′, reverse 5′-TGTTTGTGGTGATGTCGCTTC-3′ *Gapdh*: forward 5′-GGACCTCATGGCCTACATGG-3′, reverse 5′-TAGGGCCTCTCTTGCTCAGT-3′. *ARK5*: forward 5′-GTTTTCTGGCCGAGTGGTTG-3′, reverse 5′-ATCGTACAGCTCCCCTTTGC-3′. *ACTA2*: forward 5′-CGTGGCTATTCCTTCGTTAC-3′, reverse 5′-TGCCAGCAGACTCCATCC-3′. *PDGFRβ*: forward 5′-GCCCTTATGTCGGAGCTGAAGA-3′, reverse 5′-GTTGCGGTGCAGGTAGTCCA-3′. *COL1A1*: forward 5′-CAGCCGCTTCACCTACAGC-3′, reverse 5′-TCAATCACTGTCTTGCCCCA-3′. *PAI-1*: forward 5′-AGTGGACTTTTCAGAGGTGGA-3′, reverse 5′-GCCGTTGAAGTAGAGGGCATT-3′. *TIMP1*: forward 5′-TGTTGTTGCTGTGGCTGATAGC-3′, reverse 5′-TCTGGTGTCCCCACGAACTT-3′. *TβRI*: forward 5′-ACGGCGTTACAGTGTTTCTG-3′, reverse 5′-GCACATACAAACGGCCTATCT-3′. *SMAD4*: forward 5′-AAAGGTGAAGGTGATGTTTGGGTC-3′, reverse 5′-CTGGAGCTATTCCACCTACTGATCC-3′. *SMURF1*: forward 5′-GGTGGCACTGCACTCCTAGAAC-3′, reverse 5′-GCGCGGACCCAAAGTAGAAC-3′. *SMURF2*: forward 5′-ATGCCTGACAGTCCCAAGGT-3′, reverse 5′-ACCTGCCTGAGGCTGTTGTT-3′. *E-Cadherin*: forward 5′-CATGAGTGTCCCCCGGTATC-3′, reverse 5′-CAGTATCAGCCGCTTTCAGA-3′. *Vimentin*: forward 5′-ATTCCACTTTGCGTTCAAGG-3′, reverse 5′-CTTCAGAGAGAGGAAGCCGA-3′. *IL-1β*: forward 5′-AACAGCGAGGGAGAAACTGG-3′, reverse 5′-GGTCGGAGATTCGTAGCTGG-3′. *IL-6*: forward 5′-TGCAATAACCACCCCTGACC-3′, reverse 5′-GTGCCCATGCTACATTTGCC-3′. *IL-18*: forward 5′-GGCACAAACTTTCAGAGACAGCAG-3′, reverse 5′-GTTTCTTCCTGGCTCTTGTCCTAG-3′. *IL-13*: forward 5′-TTTGGTGGCCATGGGGGATA-3′, reverse 5′-TCTGGGTGATGTTGACCAGC-3′. *TNF-α*: forward 5′-TGGGATCATTGCCCTGTGAG-3′, reverse 5′-GGTGTCTGAAGGAGGGGGTA-3′. *TGF-β1*: forward 5′-GGCCAGATCCTGTCCAAGC-3′, reverse 5′-GTGGGTTTCCACCATTAGCAC-3′. *GAPDH*: forward 5′-TGCACCACCAACTGCTTAGC-3′, reverse 5′-GGCATGGACTGTGGTCATGAG-3′.

### 4.6. Western Blotting and Immunoprecipitation

Proteins were extracted from liver tissues and the indicated cells with RIPA lysis buffer (Beyotime, Guangzhou, China). The protein concentration was measured by the BCA protein assay kit (Beyotime, Guangzhou, China). Proteins were separated using a sodium dodecyl sulfate polyacrylamide gel electrophoresis (SDS-PAGE) system and electrophoretically transferred onto polyvinylidene difluoride membranes (Millipore, Boston, MA, USA). Then membranes were incubated with primary antibodies and a horseradish peroxidase-conjugated secondary antibody respectively. The following antibodies were used for Western blotting: ARK5 (Santa Cruz, CA, USA, sc-271828), GAPDH (Santa Cruz, CA, USA, sc-47724), α-SMA (Invitrogen, Waltham, MA, USA, PA5-18292), Collagen I (Invitrogen, Waltham, MA, USA, PA5-29569), PAI-1 (Abcam, Cambridge, UK, ab182973), p-Smad2/3 (Abcam, Cambridge, UK, ab254407), Smad2/3 (Abbkine, California, USA, ABP52462), TβRI (Abcepta, Suzhou, China, AP7822c), Smad4 (Cell Signaling Technology, Beverly, MA, USA, 46535S), Smurf2 (Santa Cruz, CA, USA, sc-518164), E-cadherin (Cell Signaling Technology, Beverly, MA, USA, 3195), Vimentin (Cell Signaling Technology, Beverly, MA, USA, 5714). Protein bands were visualized with a chemiluminescent (ECL) system (Bio-Rad, Hercules, CA, USA). For immunoprecipitations, LX-2 cells were lysed in ice-cold IP Lysis Buffer (Thermo Scientific, Waltham, MA, USA) for 30 min, followed by centrifugation at 17,000× *g* for 20 min. The protein lysates were incubated with specific antibodies for 16 h at 4 °C. Then, the Dynabeads Protein G beads (Thermo Scientific, Waltham, MA, USA) was added and incubated at 4 °C for 6 h. After being washed five times with lysis buffer, denaturing eluted, proteins were analyzed by Western blotting, as described previously.

### 4.7. Cell Apoptosis Analysis

LX-2 cells were harvested and washed three times in cold phosphate-buffered saline (PBS), the supernatant was discarded after the final centrifugation, and cells were resuspended by adding 500 µL 1 × Annexin V binding buffer. Then 5 µL Annexin V-PE and 5 µL 7-Amino Actinomycin D (Elabscience, Wuhan, China) were added and incubated at room temperature for 15 min in dark. The apoptosis rate was analyzed and calculated by flow cytometry according to the manufacturer’s instructions.

### 4.8. Wound-Healing Assay

LX-2 cells were cultured in the 6-well plate and transfected with siControl or siARK5. After 24 h, straight lines were delineated at the bottom of the 6-well plate and replaced the medium. The width of scratch area was respectively measured after 0 h and 24 h, and the migration ability was analyzed by Image J 1.52 software.

### 4.9. Transwell Assay

Cell migration assays were performed using Transwell chambers (3422 corning) according to the manufacturer’s instructions. LX-2 cells (1 × 10^5^ cells/well) were seeded into the upper chamber containing 200 μL of serum-free medium, and the lower chambers were filled with 500 µL medium containing 10% FBS. After incubation for 16 h, the cells that went through the compartment were fixed with 4% paraformaldehyde and stained by Hoechst 33342 (Thermo Scientific, Waltham, MA, USA). Finally, the cells were counted under the fluorescence microscope. 

### 4.10. ELISA

LO2 cells were transfected with ARK5 overexpression plasmids. After 24 h, the supernatant was discarded and the cells were rinsed with PBS, and new medium was added to continue the culture. After 24 h, the supernatant was harvested and used for subsequent ELISA determination. For hydroxyproline detection, the liver tissue of mice was put into PBS (at a ratio of 0.1 mg/μL) for homogenization, followed by centrifugation at 3000 rpm for 10 min. The supernatant was collected for ELISA determination. IL-6 and IL-18 ELISA kits were purchased from R&D Systems. The TGF-β1 ELISA kit was purchased from Multi Sciences (Hangzhou, China). The hydroxyproline ELISA kit was purchased from Meibiao Biotechnology Co. (Jiangsu, China). All tests were carried out according to the manufacturer’s instructions.

### 4.11. Statistical Analysis

Statistical data are expressed as the mean ± standard deviations (SD). Comparisons were evaluated with the two-tailed Student’s *t* test and ANOVA. *p* < 0.05 was considered statistically significant. Statistical analyses were conducted with the SPSS 12.0 software.

## 5. Conclusions

Our study found that ARK5 expression was elevated in fibrotic liver tissue and was correlated with disease severity. ARK5 is highly expressed in both HSCs and hepatocytes, regulates HSCs activation, hepatocyte EMT, and induces the expression and secretion of inflammatory factors in hepatocytes. ARK5 in HSCs and hepatocytes synergistically promoted the development of liver fibrosis (Figure 8F). Our data suggest that ARK5 may be a potential therapeutic target for liver fibrosis and provides a novel mechanism for the development of liver fibrosis.

## Figures and Tables

**Figure 1 ijms-23-13084-f001:**
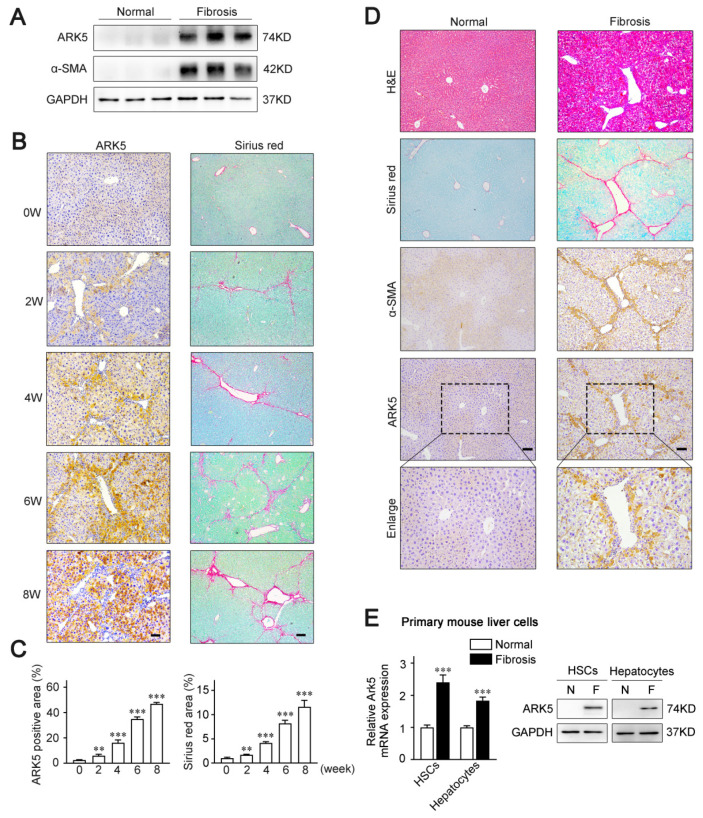
ARK5 associated with liver fibrosis progress and mainly increased in activated hepatic stellate cells and hepatocytes. (**A**) Western blot analysis of ARK5, α-SMA expressions in liver tissues from normal and CCl_4_-treated mice (*n* = 3). (**B**) Immunohistochemistry analysis of ARK5 and Sirius red staining in indicated groups of CCl_4_-induced fibrosis models. Scale bar, 100 μm. (**C**) Quantification of ARK5 immunohistochemistry (Left) and Sirius red staining (Right). (**D**) Hematoxylin-eosin (H&E) staining, Sirius red staining, and immunohistochemistry analysis of α-SMA and ARK5 of serial sections of mouse normal or fibrotic mouse liver tissues. Scale bar, 100 μm. (**E**) ARK5 mRNA and protein expression in primary mouse HSCs and hepatocytes isolated from normal or CCl_4_-induced mouse fibrotic livers. Mean ± SD (*n* = 3 per group), ** *p* < 0.01, *** *p* < 0.001 versus normal group. α-SMA, α-smooth muscle actin.

**Figure 2 ijms-23-13084-f002:**
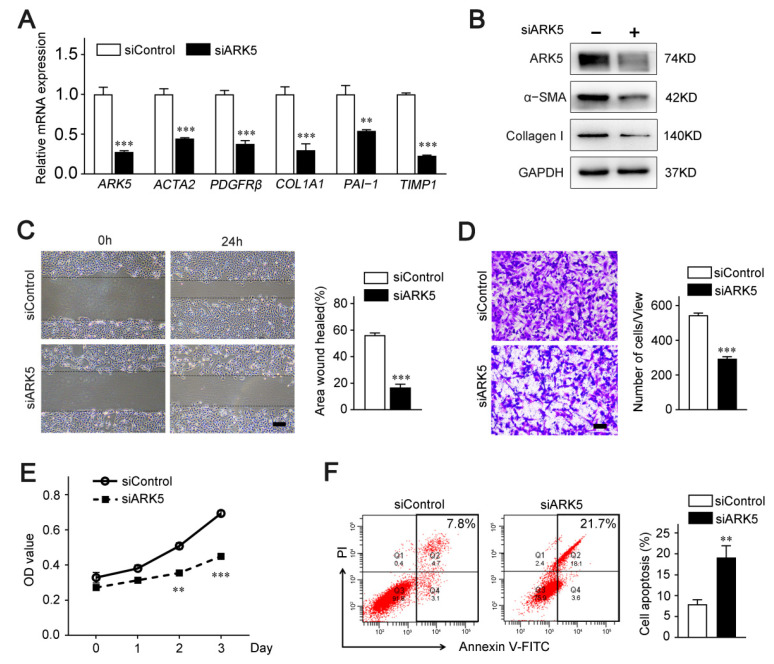
ARK5 promotes HSCs activation, proliferation and survival. (**A**) Quantitative PCR analysis of ARK5 and HSC activation-related genes in the LX-2 cells transfected with siControl or siARK5. (**B**) The expression of ARK5, α-SMA, Collagen I was detected by Western blotting in the LX-2 cells transfected with siControl or siARK5. (**C**) The migration, proliferation and survive capability of LX-2 cells transfected with siControl or siARK5 was measured using the wound-healing assay. Scale bar, 100 μm. (**D**) The migration of LX-2 cells transfected with siControl or siARK5 was analyzed by the transwell assay. Scale bar, 200 μm. (**E**) The proliferation capability of LX-2 cells transfected with siControl or siARK5 was measured using the CCK8 assay. (**F**) LX-2 cells were transfected with siControl or siARK5, and the apoptotic cells was examined by flow cytometry. Mean ± SD (*n* = 3 per group), ** *p* < 0.01, *** *p* < 0.001 versus siControl group. ACTA2, actin alpha 2; PDGFRβ, platelet-derived growth factor receptor-β; COL1A1, collagen type I alpha 1; PAI-1, plasminogen activator inhibitor type-1; TIMP1, tissue inhibitor of metalloproteinase 1; α-SMA, α-smooth muscle actin; V-FITC, annexin-V fluorescein isothiocyanate.

**Figure 3 ijms-23-13084-f003:**
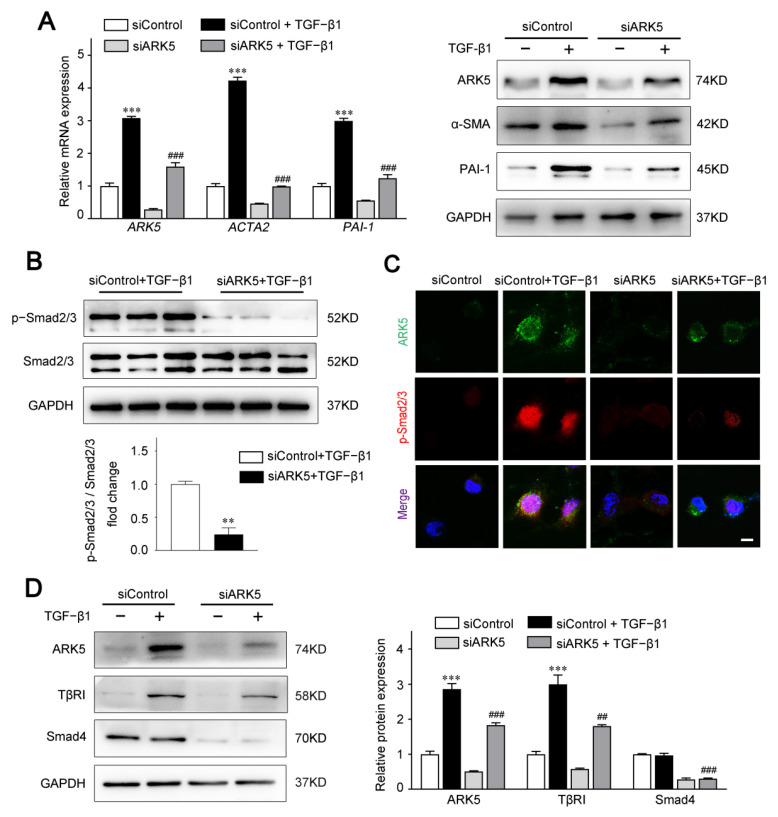
Knockdown of ARK5 expression attenuates the TGF-β-Smad2/3 signaling pathway. (**A**) Quantitative PCR and Western blot for ARK5 and fibrotic genes in control and ARK5-knockdown LX-2 cells treated with or without TGF-β1 (10 ng/mL) for 48 h. (**B**) Representative Western blot for phosphorylated and total Smad2/3 in control and ARK5-knockdown LX-2 cells treated with TGF-β1 (10 ng/mL) for 30 min. (**C**) Immunofluorescence staining for ARK5 (green) and phosphorylated Smad2/3 (red) in LX-2 in indicated groups. Cells were treated with or without TGF-β1 (10 ng/mL) for 6 h before analysis. Scale bar, 25 μm. (**D**) Western blot for ARK5, TβRI and Smad4 in control and ARK5-knockdown LX-2 cells. Mean ± SD (*n* = 3 per group), ** *p* < 0.01, *** *p* < 0.001 versus siControl group. ^###^
*p* < 0.001, ^##^
*p* < 0.01 versus siControl + TGF-β1 group. ACTA2, actin alpha 2; α-SMA, α-smooth muscle actin; PAI-1, plasminogen activator inhibitor type-1; TβRI, Transforming growth factor-beta receptor type I.

**Figure 4 ijms-23-13084-f004:**
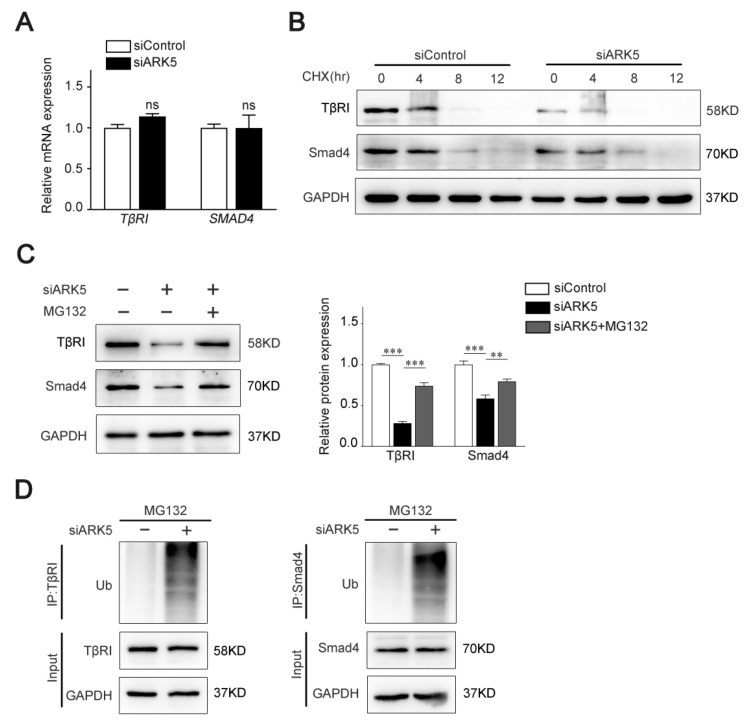
Knockdown of ARK5 promotes the degradation of TβRI and Smad4 at the protein level. (**A**) Quantitative PCR for TβRI and Smad4 in control and ARK5-knockdown LX-2 cells. (**B**) Western blot for TβRI and Smad4 in control and ARK5-knockdown LX-2 cells treated with CHX (50 μg/mL) for the indicated time. (**C**) Western blot for TβRI and Smad4 in control and ARK5-knockdown LX-2 cells treated with or without MG132 (20 μM). (**D**) Ubiquitination assays for TβRI and Smad4 ubiquitination after ARK5 knockdown in LX-2 cells. Mean ± SD (*n* = 3 per group). ns (not significant), ** *p* < 0.01, *** *p* < 0.001 versus siControl group. CHX, cycloheximide; IP, immunoprecipitation; Ub, ubiquitin.

**Figure 5 ijms-23-13084-f005:**
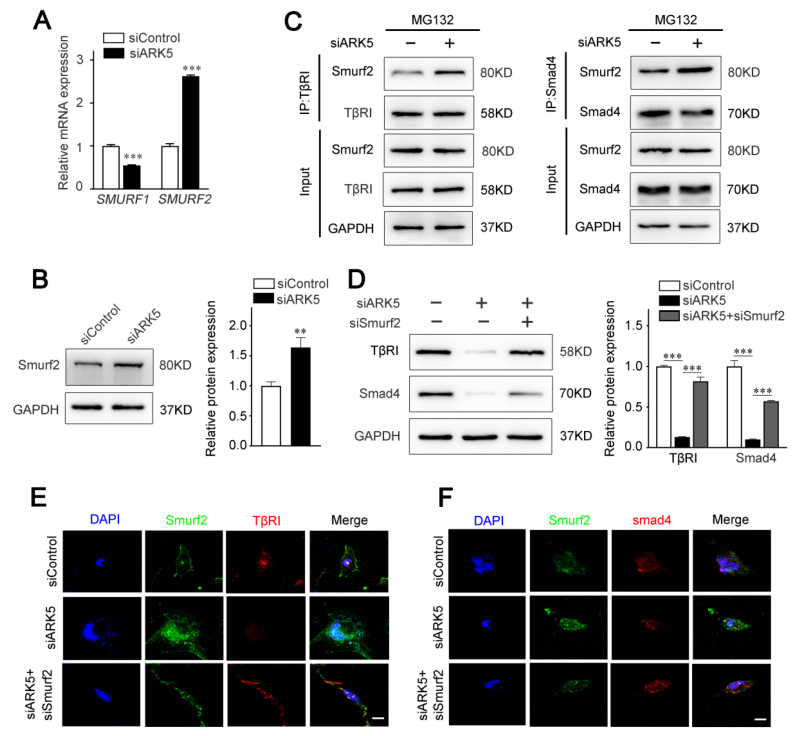
ARK5 protects TβRI and Smad4 from Smurf2 mediated degradation. (**A**) Quantitative PCR for SMURF1 and SMURF2 in control and ARK5-knockdown LX-2 cell. (**B**) Western blot for Smurf2 in control and ARK5 knockdown LX-2 cells. (**C**) Co-IP analysis of interactions between TβRI or Smad4 with Smurf2 in control and ARK5-knockdown LX-2 cells. (**D**) Western blot for TβRI and Smad4 in LX-2 cells with indicated treatments. (**E**,**F**) Immunofluorescence for Smurf2 (green) with TβRI (red) (left) and Smurf2 (green) with Smad4 (red) (right) in LX-2 cells with indicated treatments. Scale bars, 20 μm. Mean ± SD (*n* = 3 per group). ** *p* < 0.01, *** *p* < 0.001 versus siControl group.

**Figure 6 ijms-23-13084-f006:**
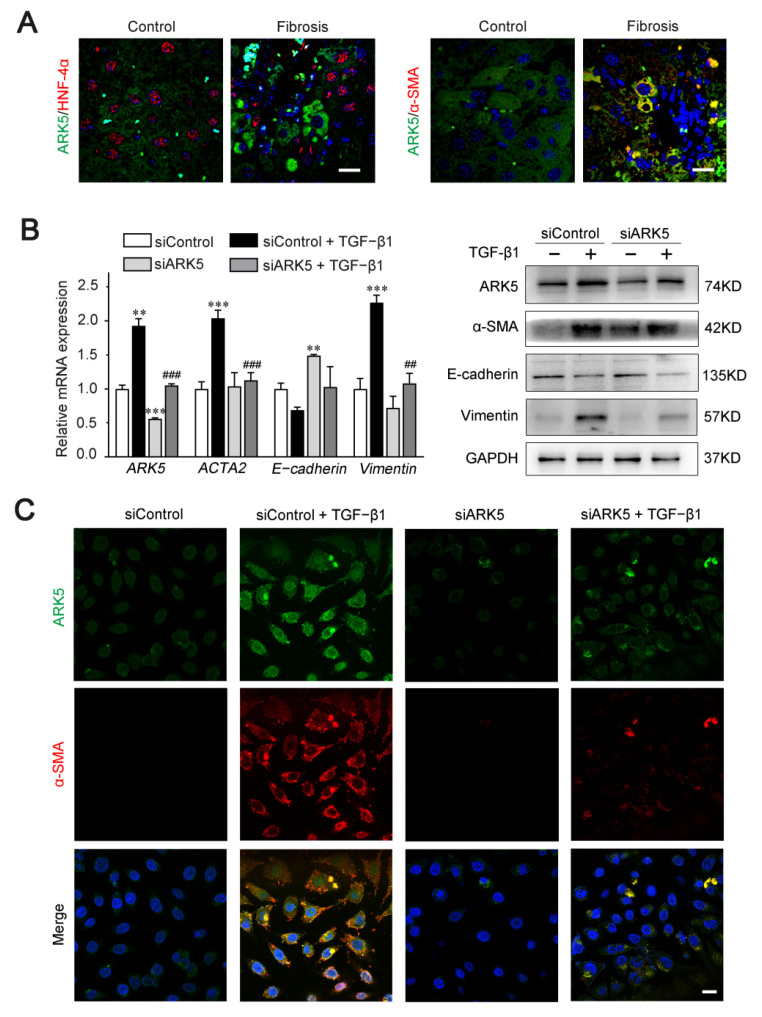
ARK5 expression was associated with hepatocyte EMT. (**A**) Immunofluorescence staining showing ARK5 (green) and HNF-4α (red) expression, ARK5 (green) and α-SMA (red) expression in the control and fibrotic livers. Scale bar, 20 μm. (**B**) The expression of ARK5, α-SMA, E-cadherin and Vimentin was detected by qPCR and Western blotting in control and ARK5-knockdown LO2 cells treated with or without TGF-β1 (10 ng/mL) for 48 h. (**C**) Immunofluorescence staining showing ARK5 (green) and α-SMA (red) expression in control and ARK5-knockdown LO2 cells treated with or without TGF-β1. Scale bar, 20 μm. Mean ± SD (*n* = 3 per group). ** *p* < 0.01, and *** *p* < 0.001 versus siControl group. ^##^
*p* < 0.01, ^###^
*p* < 0.001 versus siControl + TGF-β1 group. ACTA2, actin alpha 2; HNF-4α, Hepatocyte nuclear factor 4-alpha.

**Figure 7 ijms-23-13084-f007:**
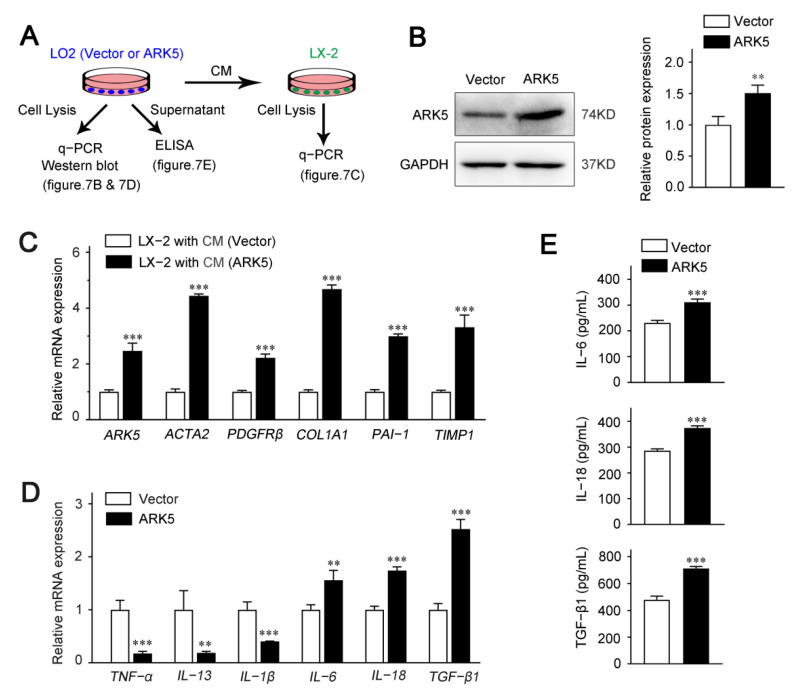
Up-regulation of ARK5 in hepatocytes promotes HSCs activation via crosstalk. (**A**) Schematic of experimental design showing LX-2 was cultured in medium from LO2 cells transfected with ARK5 overexpression vector and plasmid. Supernatant was collected for ELISA and cells were harvested for PCR. CM, conditioned medium. (**B**) Western blot analysis of ARK5 protein expression in LO2 cells transfected with ARK5 expression vector and plasmid. (**C**) Quantitative PCR analysis of ARK5 and HSC activation-related genes in LX-2 cells cultured with CM. (**D**) Quantitative PCR analysis of inflammation-related genes in LO2 cells transfected with ARK5 expression vector and plasmid. (E) Protein levels of IL-6, IL-18 and TGF-β1 in the supernatant were detected by ELISA. Mean ± SD (*n* = 3 per group). ** *p* < 0.01, and *** *p* < 0.001 versus vector group. CM, conditioned medium; ELISA, enzyme-linked immunosorbent assay.

**Figure 8 ijms-23-13084-f008:**
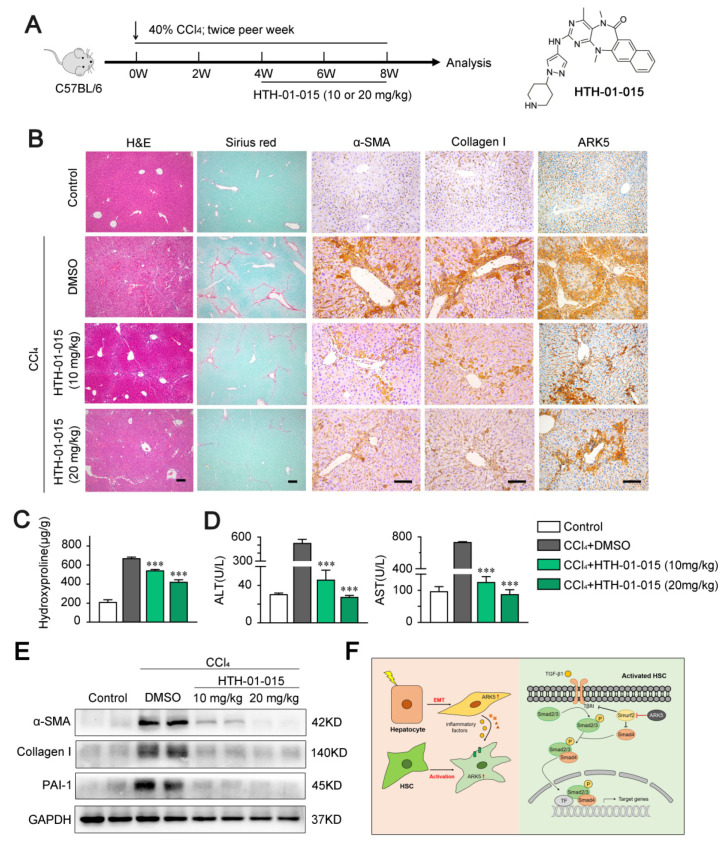
HTH-01-015 alleviates liver fibrosis in a CCl_4_-induced mouse model. (**A**) Schematic of the experimental design of HTH-01-015 (10 or 20 mg/kg) treatment in a CCl_4_-induced liver fibrosis mouse model. (**B**) Representative images of mouse livers stained with H&E, Sirius red, α-SMA, Collagen I and ARK5. Scale bars, 50 μm. (**C**) Hydroxyproline content in liver tissues of indicated groups. (**D**) Levels of ALT, AST in serum from mice. (**E**) Western blot analysis showing α-SMA, Collagen I and PAI-1 protein levels of mouse liver tissues in indicated groups. (**F**) Schematic of ARK5 promoting liver fibrosis. Mean ± SD (*n* = 3 per group). *** *p* < 0.001 versus CCl_4_ + DMSO group. DMSO, dimethyl sulfoxide; ALT, alanine aminotransferase; AST, aspartate aminotransferase.

## Data Availability

The data that support the findings of this study are openly available on request from the corresponding author.

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
