# Peer review of "Enhanced Expression of ARK5 in Hepatic Stellate Cell and Hepatocyte Synergistically Promote Liver Fibrosis"

_ijms, 2022, doi:10.3390/ijms232113084_

Round 1
Reviewer 1 Report (Previous Reviewer 1)
Authors have addressed concerns
Author Response
Thank you for your precious time to review our articles, I would like to express my high respect to you.
Reviewer 2 Report (New Reviewer)
The authors have endeavored to elucidate the role of ARK5 in hepatic fibrosis in both hepatic and stellate cells and also elucidated the cross-talk between the cells. Although interesting, there are some remarks that need to be addressed which will clarify and strengthen the manuscript.
General remarks:
Overall, the authors confirmed almost all data on protein level with western blot or other techniques. However, when the bands are largely similar, quantification and statistical analyses will be required to make certain statements. For example: Line 152-154 ‘However, proteasome inhibitor MG132 treatment reversed the downregulated of TβRI and Smad4 proteins by ARK5 knockdown in LX-2 cells (Figure 4C)’ which is not apparent from the picture. Line 170 stated that there is a ‘significant’ increase but without quantification or statistical test. This holds true throughout the manuscript.
Major comments
· Line 120-122 states that ‘suggesting a pivotal role of ARK5 in the survival of HSCs’ however the mouse isolated HSCs do not show any ARK5 and appear to be fully functional. Would the inhibition of ARK5 in the isolated healthy and fibrotic mouse HSC have an impact on HSC survival? Furthermore, in Fig S1, the cells are treated with the SB431542 TGFBRI inhibitor and resulted in a decrease of ARK5. Did this cause a decrease the survival of the cells?
· In Fig 2C it is claimed that the cells have a reduced capacity of migration, however, they also have less proliferation and increased apoptosis. Can it reliably be said that the reduction in wound closure is due to a lack of migration and not just to a lack of cells?
· In Fig 7 and in Line 208-210 the point is made that ´Taken together, these findings suggested that hepatocyte ARK5 promotes HSCs activation and liver fibrosis by regulating the release of TGF-β1’. However, the ARK5 expression in the LX-2 cells is missing. Also, the increase in TGFB1 protein in the CM seems very limited, there is an increase of 400 to 600 pg/ml with the introduction of the ARK5 overexpression plasmid. While the increase in the TGFB1 in the CM is limited, the effect on the LX-2 cells is very similar as when treated with 10 ng/ml TGFB1 in Fig 3A, suggesting that not TGFB1 or at least not TGFB1 alone is the cause of the increase in gene expression. This needs comments in the manuscript.
· Although the results in Fig 8 look promising there is no measurement of ARK5. As demonstrated in Fig 1 there is no analysis on whether the inhibitors affect ARK5 expression in the mouse liver. Also, the separate isolation of the HSC and hepatocytes could reveal valuable information regarding the cell types affected by the inhibitor.
Minor comments
· Fig. 1A The legend states that there is a n=7 but only 3 samples are shown per condition and there is no quantification. Please include all data points.
· Fig. 1B The ARK5 positive area in the week 8 samples seems quite big compared to the control but the quantification only goes up to about 50%. Please clarify.
· Naming inconsistencies throughout the text. Example: ‘epithelial-mesenchymal transition’ line 27 or ‘epithelial to mesenchymal transition’ line 51
· Experiment Fig 1E and Fig 2A have differently labeled axis ‘relative’ and ‘fold change’ but show the same type of data, please make them consistent.
· Typo line 72
· Line 179 EMT already abbreviated
Round 2
Reviewer 2 Report (New Reviewer)
Point 2.
The conclusion that ARK5 is required for survival does not comply with the results that the mouse isolated HSCs do not express ARK5 and are healthy.
In the absence of available mouse HSCs, treatment of LX-2 cells with HTH-01-015015 would add further support to the claim that ARK5 function is required for activated HSC survival. The link is at present not shown in a satisfactory way.
Since the authors state that ‘HTH-01-015015 is a highly selective protein kinase inhibitor, which mainly affects ARK5 by inhibiting the phosphorylation of myosin phosphatase target subunit 1 (MYPT1), a substrate of ARK5. It does not directly inhibit ARK5 protein expression.’
Point 3.
Then the text should be rephrased to reflect the ‘significant decrease in wound healing which is a process that combines proliferation, survival and migration’ instead of a directly claiming ‘significant decrease in migration’
Point 4.
The authors claim to only highlight activation markers in Fig 7C because they have sufficiently demonstrated that ARK5 is expressed in LX-2 cells. However, the point of the figure is to see whether the activation (overexpression) of ARK5 in the hepatocytes can induce activation of LX-2 cells. As they state themselves, LX-2 are already in an activated state but can be activated further according to the data. In figure S1A the authors have demonstrated that ARK5 is a TGFB1 responsive gene and that TGFB1 can increase the ARK5 expression in LX-2 cells. Therefore, the authors must show whether the ARK5 expression in LX-2 is also increased using ARK5 overexpression CM.
NOTE: Fig S1A is incorrectly labelled and states siCTRL vs siARK5 which should be control vs TGFB1 treatment
The authors have added a section in the discussion to nuance the results obtained in figure 7 but the result section still claims that TGFB1 is the sole driver of this process. The data does not sufficiently support this, and the result section should be adjusted on this point.
Point 5.
Please include the ARK5 staining in the manuscript
Round 3
Reviewer 2 Report (New Reviewer)
The manuscript is now acceptable for publication
This manuscript is a resubmission of an earlier submission. The following is a list of the peer review reports and author responses from that submission.
Round 1
Reviewer 1 Report
The author's present data from well designed and constructed experiments to demonstrate the role of ARK5 in liver fibrosis. The studies are primarily conducted with cell lines such as aHSC (LX-2) cells and hepatocytes with complementary data from mouse models of fibrosis (CCL4). The only request I would have of the authors is perhaps if they have any human data (such as ihc) to further support their study.
Reviewer 2 Report
You et al. describe that ARK5 is a critical driver of liver fibrosis. Their data show that ARK5 is upregulated during liver fibrosis and prevents degradation of TGFbeta receptor 1 increasing TGFbeta signaling. This can drive HSC activation and hepatocyte EMT. The use of HTH-01-015 attenuated measures of fibrosis giving support of targeting ARK5 as a potential therapeutic. While the study is well-designed and has much merit due to dissecting ARK5 interactions during liver fibrosis, there is a significant concern I have with this study, which dampens my enthusiasm of this work.
1) A significant problem with this manuscript lies in the blot images. The whole blot images provided in supplemental data are obviously cropped and not true whole blot images. In addition, there are additional issues like in figure 3B. SMAD2 is molecular weight 60 while SMAD3 is at molecular weight 52 (with no change based on phosphorylation). The authors show phosphorylated SMAD2/3 at 60kDa and total SMAD2/3 at 52kDa. There should be 2 bands in each if truly showing SMAD2/3. With the whole blot images being cropped, I could not determine what bands were actually being quantified for the graph in figure 3B. So to summarize how these problems need to be fixed, the authors must provide full blot images and correct their SMAD2/3 immunoblot images and quantification throughout their results.
Round 2
Reviewer 2 Report
The authors have addressed my concerns. That being said, I personally do not think it is appropriate to combine blots in the manner described in the response to reviewers. Due to this reason, I cannot recommend this study for publication.